# Overexpression of a *Malus baccata* NAC Transcription Factor Gene *MbNAC25* Increases Cold and Salinity Tolerance in *Arabidopsis*

**DOI:** 10.3390/ijms21041198

**Published:** 2020-02-11

**Authors:** Deguo Han, Man Du, Zhengyi Zhou, Shuang Wang, Tiemei Li, Jiaxin Han, Tianlong Xu, Guohui Yang

**Affiliations:** Key Laboratory of Biology and Genetic Improvement of Horticultural Crops of Northeast Region, Ministry of Agriculture and Rural Affairs, College of Horticulture & Landscape Architecture, Northeast Agricultural University, Harbin 150030, China; deguohan_neau@126.com (D.H.); duman102@163.com (M.D.); zhouzhengyi0101@163.com (Z.Z.); ws18045312966@163.com (S.W.); A02140301@163.com (J.H.); 18846829610@163.com (T.X.)

**Keywords:** *Malus baccata* (L.) Borkh, *MbNAC25*, cold stress, salt stress

## Abstract

NAC (no apical meristem (NAM), Arabidopsis thaliana transcription activation factor (ATAF1/2) and cup shaped cotyledon (CUC2)) transcription factors play crucial roles in plant development and stress responses. Nevertheless, to date, only a few reports regarding stress-related NAC genes are available in *Malus baccata (L.)* Borkh. In this study, the transcription factor *MbNAC25* in *M. baccata* was isolated as a member of the plant-specific NAC family that regulates stress responses. Expression of *MbNAC25* was induced by abiotic stresses such as drought, cold, high salinity and heat. The ORF of *MbNAC25* is 1122 bp, encodes 373 amino acids and subcellular localization showed that MbNAC25 protein was localized in the nucleus. In addition, *MbNAC25* was highly expressed in new leaves and stems using real-time PCR. To analyze the function of *MbNAC25* in plants, we generated transgenic *Arabidopsis* plants that overexpressed *MbNAC25.* Under low-temperature stress (4 °C) and high-salt stress (200 mM NaCl), plants overexpressing *MbNAC25* enhanced tolerance against cold and drought salinity conferring a higher survival rate than that of wild-type (WT). Correspondingly, the chlorophyll content, proline content, the activities of antioxidant enzymes superoxide dismutase (SOD), peroxidase (POD) and catalase (CAT) were significantly increased, while malondialdehyde (MDA) content was lower. These results indicated that the overexpression of *MbNAC25* in *Arabidopsis* plants improved the tolerance to cold and salinity stress via enhanced scavenging capability of reactive oxygen species (ROS).

## 1. Introduction

Plants play a very important role in our life, but various stresses in the environment affect the normal growth and development of plants seriously. The stress signal is perceived and transduced, ultimately resulting in the expression of functional proteins that protect the plant. Stress-responsive genes are mainly regulated by transcription factors, which are particularly important in the tolerance of plants to changeable environmental conditions [1]. NAC genes play important roles in plant resistance including abiotic and biotic stress responses. Previous studies have shown that there are a large number of transcription factors (TFs) exist in plants, which affect plants’ response to stress [2]. NAC TFs (NAM, ATAF1/2 and CUC2) are widely distributed in plants. So far, 117 NAC family TFs have been found in the *Arabidopsis* genome, 151 in rice, 79 in grape, 163 in poplar, 152 in soybean and at least 152 in tobacco [3,4,5,6,7].

The proteins encoded by NAC family genes form homodimers or heterodimers at the N-terminal, consist of about 150 amino acids. The N-terminal regions of NAC proteins share a conserved DNA binding region, which is divided into five subdomains (A–E) [8], whereas the C-terminal region that contains a transcriptional regulatory domain is highly diversified. 

The regulatory mechanism of NAC TFs consists of two types. The first one is the regulation of transcriptional levels including phosphorylation and ubiquitination miRNAs, which can regulate the expression of TFs at the protein level. The other is that post-transcriptional regulation can be carried out by binding to the target mRNA. NAC transcription factors can regulate target genes by binding to NACRS (NAC recognition sequence) or NACBS (NAC binding sequence), thereby affecting fruit appearance, ripeness, aging, flavor and nutritional quality [9,10]. In addition, NAC transcription factors also bind DNA or other protein kinases to regulate growth and development in plants and various physiological processes, including top meristem [11], secondary wall formation [12], flower organ development [13], bud differentiation [14], embryo development [15], lateral root formation [16,17], plant organ aging and fruit maturity [18].

Abiotic stress factors such as drought, high salt, cold and low temperature seriously affect the growth of plants. Under these stresses, a large number of reactive oxygen species will be generated among plant cells. In this case, superoxide dismutase (SOD), CAT (catalase), peroxidase (POD) and other enzyme activities will increase to a certain extent to remove harmful substances in the plant and maintain the normal development of the plant [19]. NAC TFs enhance stress resistance by regulating physiological and biochemical processes in plants. So far, the NAC TF family has been accurately analyzed in a variety of plants. Two members of the NAC family in soybean, *GmNAC20* and *GmNAC11*, regulate the development of lateral roots by regulating DREBs (dehydration responsive element binding protein) and other transcription factors related to external stress to improve salt tolerance of plants [20]. Overexpression of *TaNAC2* from wheat increases the expression of stress-resistant genes in the *Arabidopsis* and related physiological parameters of stress, thereby increasing the resistance of plants [21]. Zheng found that overexpression of *OsNAC045* in transgenic rice significantly improved drought and salt tolerance of rice [22]. Under high light, *ANAC078* in *Arabidopsis* responds to high light and high-temperature stresses, and regulates the synthesis of flavonoids, allowing plants to accumulate anthocyanins under high light. Under salt stress conditions, overexpression of *RhNAC31,* a NAC TF in *Rosa hybrida*, displayed increased germination rates and lower levels of H_2_O_2_, malondialdehyde (MDA), peroxidase (POD) and superoxide dismutase (SOD). Moreover, its overexpression enhanced cold tolerance in *Arabidopsis*, conferring a higher survival rate and reduced reactive oxygen (H_2_O_2_ and O^2−^) levels [23]. In the process of plant response to stress, some NAC transcription factors also have a negative regulatory effect on plant response to stress. *Glycine max* NAC-like gene 2 (GmNAC2) functions as a negative regulator during abiotic stress, and participates in reactive oxygen species (ROS) signaling pathways through modulation of the expression of genes related to ROS scavenging [24].

However, the roles of NAC genes in *Malus* plant stress responses are not well known. *Malus baccata* is widely used as an apple rootstock ornamental tree all over the world. It is highly resistant to low temperature and drought [25]. In order to understand the role of NAC gene under low-temperature and salt stress more comprehensively, a NAC transcription factor was isolated from the *Malus baccata (L.)* Borkh and named MbNAC25. The function of the transcription factor was analyzed and identified to provide new candidate genes for cold and salt tolerance breeding.

## 2. Results

### 2.1. Isolation and Phylogenetic Relationship of MbNAC25

The length of the *MbNAC25* gene is 1122 bp, encoding 373 amino acids (Figure 1A). The online software tool ProtParam was used to predict the theoretical molecular weight of this protein is predicted to be 41.488 kDa and the theoretical isoelectric point (pI) is 8.70. The average hydrophilic coefficient is −0.684, which indicated that the protein is hydrophilic. This protein contains 20 kinds of amino acids. Leu (7.8%), Lys (6.7%), Pro (9.1%), Ser (12.8%) and Thr (7.0%) are relatively abundant.

The evolutionary relationship between MbNAC25 protein and other NAC proteins from different species was analyzed by DNAMAN (v5.0) (Figure 1B). The sequences in the red frame are the conserved amino acid sequences of the *MbNAC25* gene, which consist of 143 amino acids. They are also conservative sequences in NAC TFs of other species. In addition, the phylogenetic tree shows that MbNAC25 protein has the highest homology with MdNAC25-like (NP_001280970.1, from *Malus domestica*).

### 2.2. Subcellular Localization of MbNAC25 Protein

In order to determine the specific location of MbNAC25 protein, a fusion expression vector of green fluorescent protein (GFP) with *MbNAC25* gene was constructed. As a control, the fluorescence of GFP was distributed in the plasma, membrane and nucleus (Figure 2B), while the *MbNAC25*-GFP was only distributed in the nucleus with 4′, 6-diamidino-2-phenylindole (DAPI) staining (Figure 2E). It can be concluded that the MbNAC25 protein was expressed in the nucleus.

### 2.3. Expression Analysis of MbNAC25 in M. baccata

In the control condition, the expression of the *MbNAC25* gene was higher in new leaves and stems than old leaves and roots (Figure 3A). In cold, the expression of *MbNAC25* increased rapidly, reached the maximum at 3 h in the new leaf and then decreased. Under high salinity and high-temperature stress, the expression level of *MbNAC25* increased for 12 h, then decreased gradually. For drought stress, the expression levels of *MbNAC25* increased after 12 h to 24 h of dehydration treatment then decreased (Figure 3B). Furthermore, the expression level of *MbNAC25* in the root peaked at 12 h, 6 h, 12 h and 24 h under cold, high-salt, drought stress and high-temperature treatments, and then showed a downward trend (Figure 3C). The results showed that the expression of the *MbNAC25* gene in the new leaf and root was induced by cold, high salt, drought and high temperature.

### 2.4. Overexpression of MbNAC25 in Arabidopsis Enhances Cold Tolerance

To investigate the role of *MbNAC25* in response to cold stress in plants, transgenic *Arabidopsis* lines with overexpression of *MbNAC25* under the control of the CaMV 35S promoter were generated. Among all the T_2_ generation transformed lines, the target fragments could be detected in six transformed lines (S2, S3, S4, S6, S9, S10) by RT-PCR with a wild-type (WT, Columbia-0) *Arabidopsis* line as a negative control (Figure 4A). When the T_3_ generation of transgenic lines (S3, S6, S10, randomly selected) and WT plants were cultured under control conditions, both transgenic plants and WT plants grew well, and there was no significant difference in phenotype. However, at low temperature (−4 °C), improved cold tolerance in the transgenic (S3, S6, S10) lines were observed (Figure 4B), and the survival rate of transgenic *Arabidopsis* lines were significantly higher than the WT line at 88.43% for S3, 85.52% for S6 and 87.49% for S10, compared to WT at only 24.73% (Figure 4C). When restored to control conditions, most of the transgenic plants treated by low temperature could resume growth.

In the control condition, there were no significant differences in chlorophyll content, proline content, MDA content, SOD, POD and CAT enzyme activity between transgenic *Arabidopsis* lines and the WT line. After low-temperature treatment, chlorophyll content, proline content, SOD, POD and CAT activities of *MbNAC25* transgenic lines were significantly higher than those of the WT line, while MDA content was significantly lower than that of the WT line (Figure 5). These results showed that WT plants suffered more serious membrane damage than *MbNAC25* transgenic *Arabidopsis* plants. Hence, our results suggest that overexpression of the *MbNAC25* gene could scavenge the intracellular reactive oxygen species (ROS) by increasing the enzyme activities of SOD, POD and CAT.

### 2.5. Overexpression of MbNAC25 in Transgenic Arabidopsis Increased High Salt Tolerance

To study the role of *MbNAC25* in response to high-salt stress in plants, the transgenic *Arabidopsis* lines (S3, S6, S10) and the WT line were watered by 200 mM NaCl for seven days (Figure 6A). The more yellowing leaves were observed in the WT plant than transgenic plants. When restored to normal irrigation conditions, most of the transgenic plants treated by salt stress could resume growth and the survival rate of transgenic plants was 85.62%, 90.74% and 85.75%, respectively, while that of the WT line was only 37.52% (Figure 6B).

In order to study the reasons why the transgenic lines had better appearances under high-salt stress, the chlorophyll content, proline content, MDA content, SOD, POD and CAT activities in both transgenic *Arabidopsis* lines (S3, S6, S10) and the WT line were measured before and after salt stress. These physiological indexes of *MbNAC25* transgenic *Arabidopsis* lines and the WT line under salt stress were basically the same as those of low temperature. In addition to MDA, transgenic *Arabidopsis* have higher levels of other indicators than wild-type *Arabidopsis* (Figure 7). This indicated that *MbNAC25* transgenic plants had higher ROS scavenging enzyme activities to remove more ROS and protect the membrane.

## 3. Discussion

Previous reports have confirmed that NAC TFs are involved in many plant growth and development processes such as apical meristem development, flower morphogenesis and lateral root development. In addition, NAC proteins are also involved in plant defenses’ response to abiotic and biotic stresses [22]. In this study, a *NAC* gene, named *MbNAC25*, was isolated by a homologous cloning method from the *M. baccata*. The protein encoded by the *MbNAC25* gene has the typical structural characteristics of the NAC family. It was found that the N-terminus of the MbNAC25 protein contains a highly conserved NAC domain, but the transcriptional regulatory regions at the C-terminus are diverse. The conserved sequence of MbNAC25 protein contains 157 amino acids. In addition, the average hydrophilic coefficient of MbNAC25 protein is -0.684, which indicated that MbNAC25 protein was a hydrophilic protein. According to the analysis of protein sequence, the contents of leucine (Leu), lysine (Lys), proline (Pro), serine (Ser) and threonine (Thr) were relatively high, while methionine (Met) and tryptophan (Trp) were relatively deficient. Evolutionary tree analysis showed that MbNAC25 protein had the highest homology with *Malus domestica* NAC25 protein (Figure 1). The results of subcellular localization showed that MbNAC25 protein was localized in the nucleus (Figure 2).

Increasing evidence indicates that NAC TFs are in important regulatory roles in plants in response to stresses such as drought, low temperature and high salt. Ohnishi et al. found that under control condition, it is difficult to detect *OsNAC6* in rice, but after induction by drought, high salt, low temperature and ABA, the expression of *OsNAC6* was significantly increased; moreover, overexpression of *OsNAC6* could induce the expression of many abiotic stress-related genes [26]. Similarly, drought, high salt and ABA could induce the expression of *ANAC019*, *ANAC055* and *ANAC072* in Arabidopsis and overexpression of these three genes could also improve the drought tolerance of Arabidopsis [27]. In addition, a dehydration-induced NAC protein, RD26, is involved in a novel ABA-dependent stress-signaling pathway [28]. Tran et al. found that nine *GmNAC* genes in soybean were up-regulated in shoots and roots induced by drought stress [29]. The results of this study are similar to the findings above. According to the results of real-time quantitative PCR, the highest expression of the *MbNAC25* gene in new leaves under low-temperature, high-salt, drought and high-temperature stresses were 3 h, 12 h, 24 h and 12 h (Figure 3B), while the highest expression of the *MbNAC25* gene in the roots under same abiotic stresses were 12 h, 6 h, 12 h and 24 h (Figure 3C). It could be due to the fact that the response of roots to high salinity and drought stresses is faster than that of new leaves, which is due to the fact that stresses always start from roots in high salinity and drought treatment. However, the response rate of new leaves to low-temperature and high-temperature stresses is faster than that of roots, which indicated that stress starting from new leaves in low-temperature and high-temperature treatments. Therefore, the expression of the *MbNAC25* gene is induced by low-temperature, high salinity, drought and high-temperature stresses.

A large amount of ROS is produced in the cells when plants are subjected to stress. H_2_O_2_ and O^2·–^ were detected after drought and salt stresses, and a lower amount of H_2_O_2_ and O^2–^ was observed in the roots of transgenic plants than in the WT plant [30]. Plants have gradually formed complex and delicate mechanisms to cope with oxidative stress in the process of evolution such as active oxygen scavenger enzyme system, including peroxidase (POD), superoxide dismutase (SOD) and catalase (CAT), to remove excess reactive oxygen species and free radicals in plants. NAC proteins are involved in mediating the antioxidative system under adversity stress [31]. An elevated malondialdehyde (MDA, a product of lipid peroxidation) level is used frequently as an indicator of reactive oxygen species (ROS) and associated cell membrane degradation or dysfunction. The MDA content of bluegrass leaves gradually increased with decreasing temperature. The higher the content of MDA, the weaker the plant’s frost resistance [32]. The overexpression of GmNAC085 enhanced antioxidant capacity in the transgenic plants to reduce drought-induced oxidative damage via reducing MDA content accompanied by increased activities of superoxide dismutase, catalase and ascorbate peroxidase [33]. Our study discussed the role of *MbNAC25* in plants under cold stress. The results indicated that wild-type *Arabidopsis* showed obvious wilting, while the transgenic *Arabidopsis* were normal; that is, the overexpression of the *MbNAC25* gene could significantly improve the resistance of transgenic plants under low-temperature stress (4 °C) (Figure 4). The chlorophyll content of wild-type and transgenic *Arabidopsis* decreased under low-temperature stress, but the former decreased faster. At the same time, the physiological indexes such as proline content, MDA content, SOD, POD and CAT activity of transgenic and wild *Arabidopsis* increased after low-temperature stress, which indicated that both transgenic and wild *Arabidopsis* were damaged under low-temperature stress. Compared with wild *Arabidopsis*, these physiological indexes except MDA of transgenic *Arabidopsis* increased more, and the rising of MDA content was lower (Figure 5). This indicated that the transgenic *Arabidopsis* suffered less damage under low-temperature stress. The *MbNAC25* gene significantly enhanced the tolerance of *Arabidopsis* to low-temperature stress.

Due to the high salt concentration in the plant rhizosphere, the soil water potential is lower than the water potential in the plant tissue, which makes it difficult for the plant to absorb water. The high salt concentration can also cause water infiltration in the plant, cause water loss and even cause plant death [34]. Similarly, the wild-type after salt stress treatment showed more obvious yellowing than the transgenic *Arabidopsis* (Figure 6), and the changes of chlorophyll content, proline content, MDA content, SOD, POD and CAT activity under salt stress were similar to those under low-temperature stress, but the change ranges were different (Figure 7). The *MbNAC25* gene significantly enhanced the tolerance of *Arabidopsis* to salt stress.

In a word, our results suggest *MbNAC25* was induced by low-temperature, high-salt, high-temperature and drought stress and the overexpression of *MbNAC25* in transgenic *Arabidopsis* plants enhances tolerance to cold and salinity stress.

## 4. Materials and Methods

### 4.1. Plant Material and Treatments

The tissue culture plantlets of *M. baccata* were planted in Murashige and Skoog (MS) growth medium (MS + 0.6 mg·L^−1^ 6-BA + 0.6 mg·L^−1^ IBA). After a month, the tissue culture plantlets were transplanted into rooting medium (MS + 1.2 mg·L^−1^ IBA) until white rootstocks emerged. Then, the tissue culture seedlings should be transferred to Hoagland nutrient solution with 80% relative humidity. Moreover, the nutrient solution should be replaced every 3–4 days. When the leaves were fully unfolded, seedlings were divided into four parts for salt stress (watered with 200 mM NaCl solution), low-temperature stress (placed in 4 °C growth incubator), drought stress (watered with 15% PEG solution) and high-temperature stress (placed in 38 °C growth incubator). The seedlings cultured under normal Hoagland nutrient solution were as control. After 0, 1, 3, 6, 12, 24 and 48 h for stress treatment, the plant materials were obtained and frozen with liquid nitrogen immediately, then stored in −80 °C for RNA extraction [35].

### 4.2. Isolation and Cloning of MbNAC25

An OminiPlant RNA Kit (Kangweishiji, Beijing, China) was used to extract RNA from new leaves, old leaves, root and shoot tips of hydroponic seedlings. TransScript^®^ First-Strand cDNA Synthesis SuperMix (TransGen Biotech, Beijing, China) was used to synthesize the cDNA First Strands. RNA and cDNA were assessed by 1.0% agarose gel electrophoresis. The whole sequence of *MbNAC25* was obtained by polymerase chain reaction (PCR) with primers *MbNAC25F* and *MbNAC25R*, using the first-strand cDNA of *M. baccata* as a template. The primers used in this study are shown in Table 1. The obtained DNA fragments were purified and cloned into objective vectors using the pEASY^®^-T1 Cloning Kit (TransGen Biotech, Beijing, China) and sequenced [36,37,38,39].

### 4.3. Subcellular Localization Analysis of the MbNAC25 Protein

The DNA fragmentsof *MbNAC25* was cloned into the *BamH* I and *Xba* Isites of pSAT6-GFP-N1 vector. Gold powder containing MbNAC25-GFP plasmid was injected into onion epidermal cells by gene gun method [40], and subcellular localization was carried out [41]. The fluorescence of MbNAC25-GFP was observed under a confocal microscope [42].

### 4.4. Quantitative Real-Time PCR Analysis

The expression of *MbNAC25* gene under different abiotic stress was analyzed by RT-PCR. The specific primers were shown in Table 1. The expression of the *MbNAC25* gene was detected by the Tli RNaseH Plus kit (TaKaRa, Beijing, China) according to the manufacturer’s protocol. The expression data were analyzed using the 2^−ΔΔCT^ method [43]. The *Actin* gene (NC_024251.1, *M. domestica*) was used as a reference gene.

### 4.5. Vector Construction and Agrobacterium-Mediated Arabidopsis Transformation

The primers with restriction enzyme sites BamH I and Xba I were designed, and the target fragment was amplified by PCR.The PBI121 vector and the target fragment were simultaneously digested with Xba I and BamH I enzyme, then the target fragment was connected to the vector PBI121 to construct the PBI121-MbNAC25 super expression vector. The PBI121-MbNAC25 vector was transfected into *Arabidopsis* ecotype Columbia-0 by *Agrobacterium tumefaciens* GV3101 transformation. The seeds of transformants were seeded in 1/2MS medium (containing 50 mg·L^−1^ kanamycin) [44] to selectthe transformed lines until the T_2_ generations. Seeds of the T_2_ generation of transgenic plants (S3, S6, S10, randomly selected) were sown and germinated in a nutrient soil to vermiculite ratio of 4:1 in flowerpots (diameter 10 cm) with normal management in a growth chamber at 25 ± 1 °C under a 16 hlight (50 μmol m^−2^ s^−1^) and 8 h dark regime in parallel with the wild-type (WT) seeds. Twenty-five seedlings were grown for 3 weeks with regular irrigation prior to salt stress. Salt stress experiments were conducted by watering 200 mM NaCl solution for 7 days. Then, the 25 plants of each line (S3, S6, S10 and WT) were rewatered for 6 d to calculate the survivalrate. The experiments were performed three times foreach treatment at each time point. During the wholegrowth process, all *Arabidopsis* seedlings were observedand recorded by photographing salt stress for 0 days, 7 days and rewatered for 6 days (recover). Forlow-temperature stress, 3-week seedlings were grown in the 4 °C growth incubator parallel withwild-type (WT). The experiments were also performed three times foreach treatment at each time point. During the wholegrowth process, all *Arabidopsis* seedlings were observedand recorded by photographing cold stress for 0 h, 12 h and recovered at room temperature for 24 h (Recover).

### 4.6. Determination of Related Physiological Indexes

The seeds of transgenic *Arabidopsis* lines (S3, S6, S10) and a WT line were sown in a screening medium (1/2 MS + 50 mg·L^−1^ kanamycin) and transferred to nutrient soil for two weeks. Twenty-five plants of each transgenic lines and the WT line were placed in low temperature (4 °C for 12 h), high salt (200 mM NaCl for 7 days) and normal conditions as control. Chlorophyll content was measured according to the method of Xu et al. [45]. Proline content was measured according to the method of Huang et al. [46]. POD activity was measured using the protocols described by Sharma et al. [47]. SOD activity was measured following the methods used previously [48]. CAT activity was measured following Han et al. [49]. MDA activity was measured using the protocols described by Jiang et al. [50].

### 4.7. Statistical Analysis

SPSS software was used to analyze the differences with Duncan’s multiple range tests. Statistical differences were referred to as significant when * *p* ≤ 0.05, ** *p* ≤ 0.01.

## Figures and Tables

**Figure 1 ijms-21-01198-f001:**
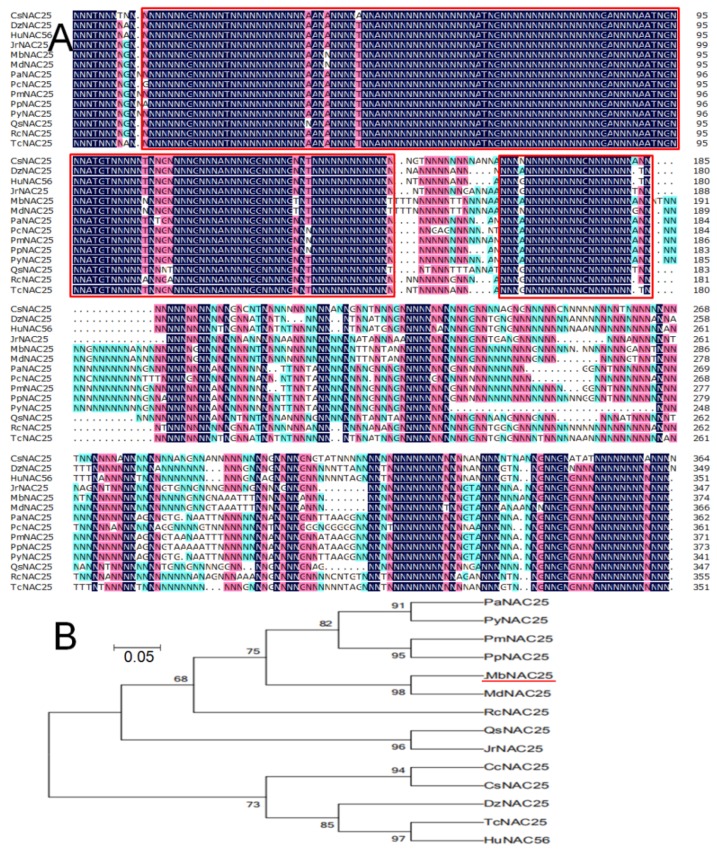
Comparison and phylogenetic relationship of MbNAC25 with other plant NAC25 transcription factors. (**A**). Homologous comparison of MbNAC25 protein with NAC25 proteins from other plant species. The sequence in the red frame is the conserved amino acid sequence. (**B**). Evolutionary tree analysis of MbNAC25 (indicated by a red line) and other plant NAC25 proteins. The accession numbers are as follows: MdNAC25-like (*Malus domestica*, NP_001280970.1), PaNAC25 (*Prunus avium*, XP_021830786.1), PyNAC25 (*Prunus yedoensis* var nudiflora, PQQ02263.1), PmNAC25 (*Prunus mume*, XP_008224995.1), PpNAC25 (*Prunus persica*, XP_007211453.1), RcNAC25 (*Ricinus communis*, ASU89555.1), QsNAC25-like (*Quercus suber*, XP_023921724.1), JrNAC25 (*Juglans regia*, XP_018807000.1), CcNAC25 (*Citrus clementina*, XP_006449287.1), CsNAC25 (*Citruus sinensis*, XP_006467845.1), DzNAC25 (*Durio zibethinus*, XP_022777366.1), TcNAC25 (*Theobroma cacao*, XP_007025712.1) and HuNAC56 (*Herrania umbratica*, XP_021294335.1).

**Figure 2 ijms-21-01198-f002:**
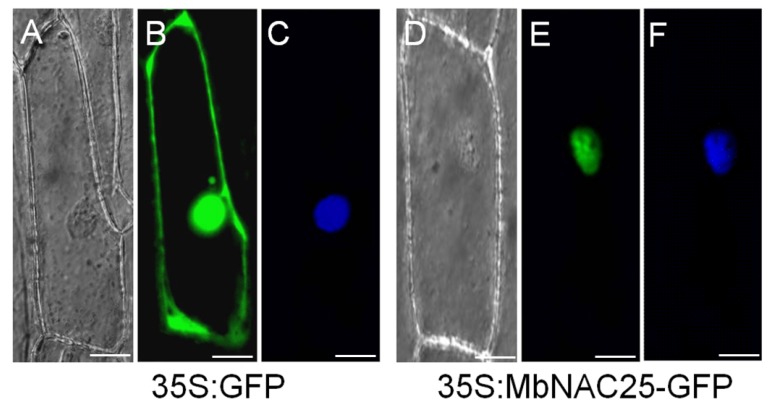
Subcellular localization of MbNAC25 protein. The 35S-GFP and 35S-MbNAC25-GFP translational products were expressed in onion epidermal cells and visualized by fluorescence microscopy in bright light (**A**,**D**); in dark field for GFP (**B**,**E**) and DAPI staining images (**C**,**F**). Scale bar corresponds to 5 μm.

**Figure 3 ijms-21-01198-f003:**
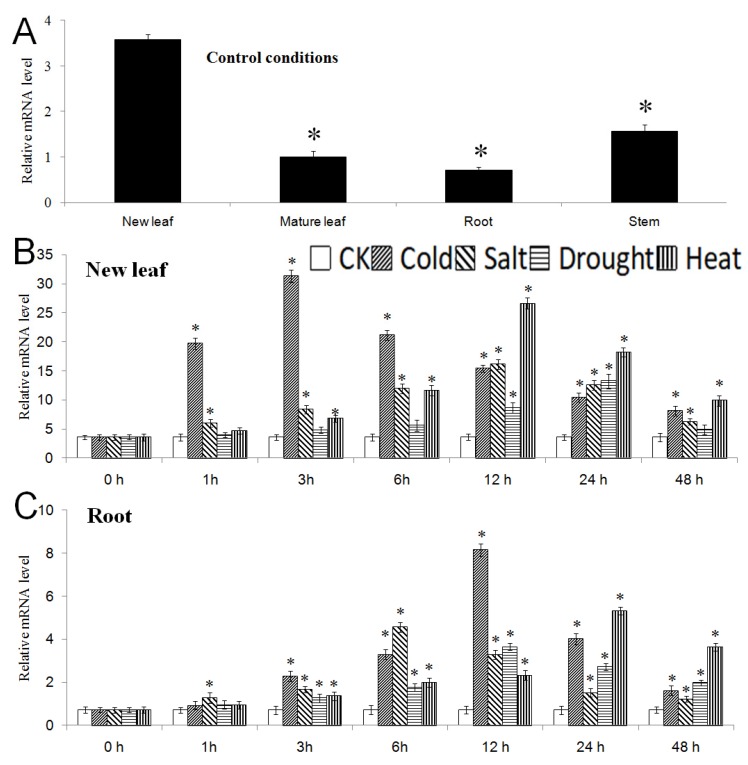
Expression of the *MbNAC25* gene in different tissues and organs of *Malus baccata.* (**A**) Expression of the *MbNAC25* gene in different organs under control condition. Asterisks above columns indicate significant difference compared to that in the new leaf (* *p* ≤ 0.05). (**B**) Expression of the *MbNAC25* gene in the new leaf and (**C**) in the root under control condition (CK), low temperature, high salt, drought and high temperature. Data represent means and standard errors of three replicates. Asterisks above columns indicate significant difference compared to that in control condition (* *p* ≤ 0.05).

**Figure 4 ijms-21-01198-f004:**
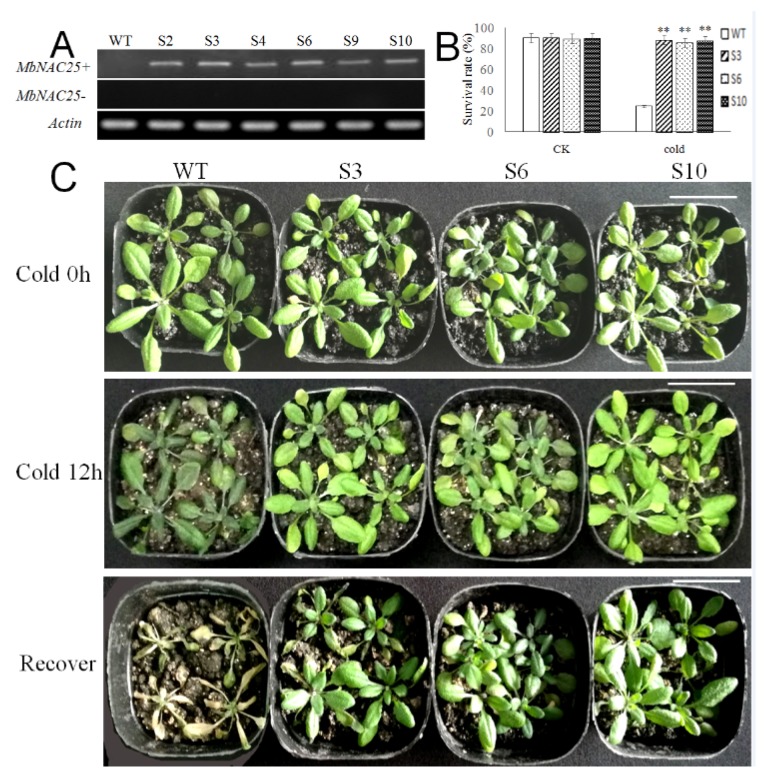
Overexpression of *MbNAC25* in *Arabidopsis* improved cold tolerance. (**A**) Expression levels of *MbNAC25* in wild-type (WT) and T_2_ transgenic *Arabidopsis* lines visualized by semi-quantitative RT-PCR using *MbNAC25* specific primer (*MbNAC25+*) and *MbNAC25* non-specific primer (*MbNAC25-*). Actin was used as control. (**B**) Survival rates of WT and transgenic lines after recovery in control condition (CK) and in cold for 12 h. The number of surviving plants was counted. Three independent experiments were performed, each with about 25 plants. Data represent means and standard errors of three replicates. Asterisks above columns indicate significant difference compared to WT (** *p* ≤ 0.01). (**C**) Phenotypes of *MbNAC25* transgenic *Arabidopsis* lines (S3, S6, S10) and WT under low-temperature stress and recovery. Scale bar corresponds to 1 cm.

**Figure 5 ijms-21-01198-f005:**
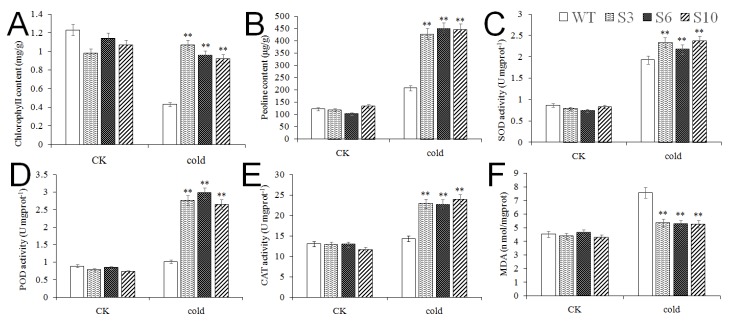
Physiological Indexes of *Arabidopsis* overexpressing of *MbNAC25* under cold stresses. (**A**) Chlorophyll content; (**B**) proline content; (**C**) superoxide dismutase (SOD) activity; (**D**) peroxidase (POD) activity; (**E**) catalase (CAT) activity; (**F**) malondialdehyde (MDA) content. Data represent means and standard errors of three replicates. Asterisks above columns indicate significant difference compared to WT (** *p* ≤ 0.01).

**Figure 6 ijms-21-01198-f006:**
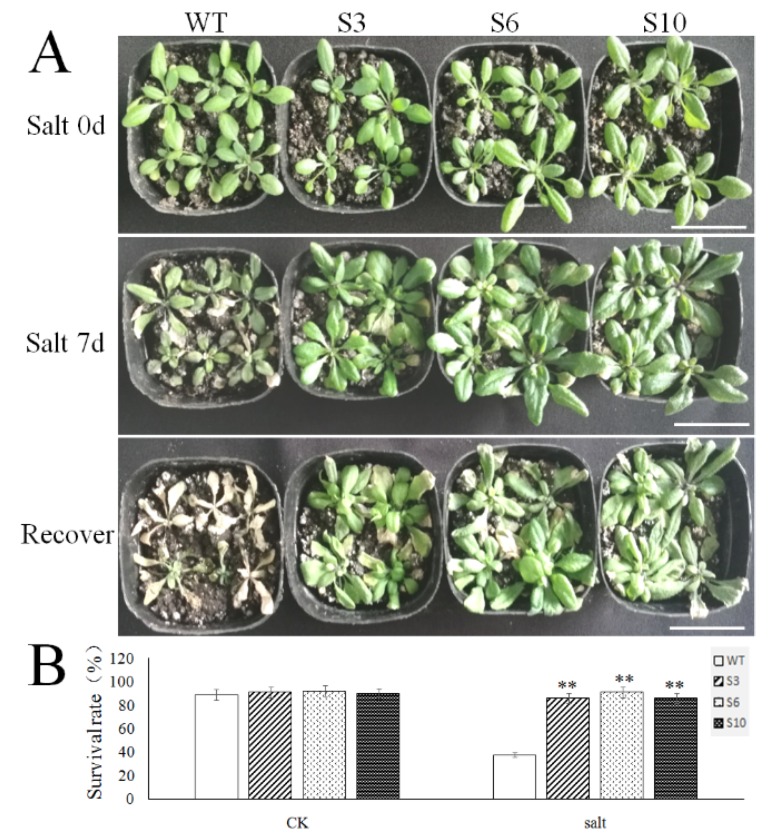
Overexpression of MbNAC25 in Arabidopsis improved salt tolerance. (**A**) Phenotypes of MbNAC25 transgenic Arabidopsis lines (S3, S6, S10) and WT under salt stress and recovery. Scale bar corresponds to 1 cm. (**B**) Survival rates of seedlings in WT and transgenic lines after recovery. Data represent means and standard errors of three replicates. Asterisks above columns indicate significant difference compared to WT (** *p* ≤ 0.01).

**Figure 7 ijms-21-01198-f007:**
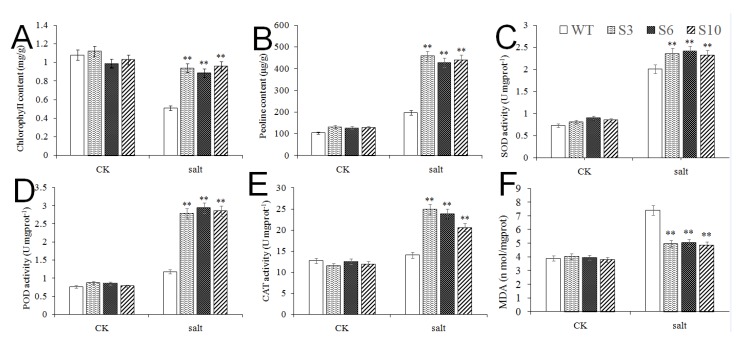
Physiological Indexes of *Arabidopsis* overexpressing of *MbNAC25* under salt stresses. (**A**) Chlorophyll content; (**B**) proline content; (**C**) SOD activity; (**D**) POD activity; (**E**) CAT activity; (**F**) MDA content. Data represent means and standard errors of three replicates. Asterisks above columns indicate significant difference compared to WT (** *p* ≤ 0.01).

**Table 1 ijms-21-01198-t001:** List of primers used in this study.

Primer Name	Primer Sequence (5′→3′)	Purpose
*MbNAC25*-F	ATGGAGAGCACAGATTCATC	full-length cDNA of *MbNAC25*
*MbNAC25*-R	CTATGAATTCCAGTTCATGCTT	full-length cDNA of *MbNAC25*
*MbNAC25*-qF	CAATGCGCAAAGGCCTACGA	qPCR
*MbNAC25*-qR	AGCAGACCCTATCGATCCCA	qPCR
Actin-F	ACACGGGGAGGTAGTGACAA	qPCR
Actin-R	CCTCCAATGGATCCTCGTTAT	qPCR

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
