# Peer review of "Overexpression of a Malus baccata NAC Transcription Factor Gene MbNAC25 Increases Cold and Salinity Tolerance in Arabidopsis"

_ijms, 2020, doi:10.3390/ijms21041198_

Round 1
Reviewer 1 Report
General comments
In this manuscript, authors have characterised the Malus baccata NAC25 transcription factor (MbNAC25) in response to abiotic stresses such as temperature and salinity. They described the gene and the protein in M. baccata and did the subcellular localization. They analysed the response of different tissues of M. baccata to several abiotic stresses and found that cold and high salt treatments highly activate the expression of MbNAC25 gene when overexpressed in Arabidopsis. They further analysed these stresses in their transgenic Arabidopsis lines by measuring physiological indexes, where they conclude that over-expression of MbNAC25 gene in Arabidopsis enhances its tolerance to cold and salt stress.
The manuscript is written well, however, the English language needs proofreading to correct spelling mistakes, singular/plural, etc. I would advise the authors to work with a writing coach or copyeditor to help them with cohesion and coherence across the manuscript. Please, be careful with the formatting, there are missing spaces across the whole manuscript.
The figures and tables illustrate correctly their results. However, authors should improve the figure legends.
Regarding the discussion, I will encourage the authors to work it a bit more. The discussion should include literature evidence that supports the conclusions the authors claim. The verbal tenses have to be adequate, I would recommend they use present tense for example, line 210-211 since they are stating information from the literature.
Materials and methods section needs more detail and English proofreading.
Bibliography, there are some references wrongly cited. Please double check all the references correspond. For example, ref 29 and 30 are wrongly cited.
Minor comments
Introduction
Why did the authors decide to study this transcription factor and why they did it in Malus baccata?
Line 37. Number of NACs in tobacco is missing here.
Line 38. “NAC family genes form homologous or heterodimers at the N…” Please correct to homodimers if corresponds.
Line 62. Authors mentioned TaNAC2 and TaNAC9, but don’t explain which species these transcription factors are from. Please mention.
Line 63. What do the authors mean with “lower reaches of Arabidopsis”?
Results
Line 73. How did the authors come up with this gene?
Line 82. What do the authors mean with “special sequences”?
Line 86. Figure 1. The formatting of the letters A and B has to be corrected. Please, make them smaller and black.
Line 89. I suggest the authors write the names of all the species that they used for the analysis. They could write it here or in the materials and methods (M&M) section as they prefer.
Line 102. Figure 2. The images lack scale bars, please add the bars and describe it also in the figure legend, such as “Scale bar corresponds to X um”.
Line 116. Figure 3. I would suggest the authors use a greyscale for the bar graphs B and C since the pattern scale is more difficult to read.
Figure 3. What do the authors mean with “Normal conditions”? Are they referring to “Control conditions”? I will recommend the authors use the term control conditions across the manuscript and that they describe those conditions in M&M. Regarding the figures 3B and 3C, what does CK stands for? The acronym is not explained.
Line 119. What did the authors compare to do the statistical analysis? Could the authors mention it in the legend? For example: “Asterisks above columns indicate significant difference compared to what, in what condition? (*P≤0.05).”. Same should be addressed in the rest of the legends.
Line 123-127. I suggest the authors move this paragraph to M&M. How they prepare the plants is not needed here.
Line 124. Authors mentioned WT as an Arabidopsis line, but they did not explain which accession they are using for their experiments. This information has to be recorded.
Line 135. The asterisks are not explained in the legend. There is no mention to the age of the plants in the experiment.
Figure 4B. How do the authors measure the survival rate? It should be described in M&M
What does CK in the axis mean? There is no explanation in the legend nor the M&M.
Line 139. Figures 4B and 4C do not correspond with the text described in the legend, please modify accordingly.
Line 147. I would recommend starting the sentence with for example “Hence, our results suggest that overexpression of…”. This connects results and hypothesis.
Line 150. “Physiological Indices” please correct to physiological indexes.
Line 153. Authors need to mention what they are comparing in the statistical analysis. Please modify across the text accordingly.
Line 156. “…were exposed to salt stress…” How do the authors perform NaCl treatment? I would suggest they use were watered if that is how they applied the stress.
Discussion
Line 192. I suggest the authors clarify this sentence, for example “… the highest homology with NAC25 in Malus domestica…”
Line 195. Do the authors mean Ohnishi et al.? Please modify the reference accordingly.
Line 196. “normal circumstances”, please explain.
Line 204. I would suggest softening the statement such as “which could be due to the fact…”
Line 218. “This study” I suggest the authors modify this to “Our study” since helps the reader focus.
Line 237. The last paragraph should be a summary with a conclusion of the work.
Materials and methods
Line 239. In the plant materials sections, there is no description of the plant materials used, for example, a description of Malus baccata variety, the Arabidopsis ecotype used, or the conditions Arabidopsis plants were grown.
This paragraph could also include the treatments applied to transgenic Arabidopsis plants on soil, cold and salt stresses.
Line 252. It will help clarify if authors mention for example “hydroponic seedlings of Malus baccata”.
Author Response
In this manuscript, authors have characterised the Malus baccata NAC25 transcription factor (MbNAC25) in response to abiotic stresses such as temperature and salinity. They described the gene and the protein in M. baccata and did the subcellular localization. They analysed the response of different tissues of M. baccata to several abiotic stresses and found that cold and high salt treatments highly activate the expression of MbNAC25 gene when overexpressed in Arabidopsis. They further analysed these stresses in their transgenic Arabidopsis lines by measuring physiological indexes, where they conclude that over-expression of MbNAC25 gene in Arabidopsis enhances its tolerance to cold and salt stress.
Point 1: The manuscript is written well, however, the English language needs proofreading to correct spelling mistakes, singular/plural, etc. I would advise the authors to work with a writing coach or copyeditor to help them with cohesion and coherence across the manuscript. Please, be careful with the formatting, there are missing spaces across the whole manuscript.
Response 1: Thank you for your kind suggestion. The manuscript has been polished by a native English speaker, but there may be some errors due to the tight schedule. We have done our best to correcte the errors one by one and listed them in the cover letter. If there are still errors, please give us another chance to correct them.
Point 2: The figures and tables illustrate correctly their results. However, authors should improve the figure legends.
Response 2: Thank you for your kind suggestion. We have re-written the figure legends, tried our best to improve the figure legends and make it easy to understand. (Line 190-200, Line 209-211, Line 226-231, Line 258-266, Line 288-291, Line 302-305)
Point 3: Regarding the discussion, I will encourage the authors to work it a bit more. The discussion should include literature evidence that supports the conclusions the authors claim.
Response 3: Thank you for your kind suggestion. We have improved the discussion, tried our best to find more literature evidence to support our conclusions. (Line 335-349, Line 359-370)
Point 4: The verbal tenses have to be adequate, I would recommend they use present tense for example, line 210-211 since they are stating information from the literature.
Response 4: Thank you for your kind suggestion. We have corrected the verbal tenses to present tense for stating information from the literatures across the manuscript.
Point 5: Materials and methods section needs more detail and English proof reading.
Response 5: Thank you for your kind suggestion. We have improved the materials and methods section, tried our best to make it more detail and easy to understand. (Line 478-511)
Point 6: Bibliography, there are some references wrongly cited. Please double check all the references correspond. For example, ref 29 and 30 are wrongly cited.
Response 6: Thank you for your kind suggestion. We have corrected the references across the manuscript to make them correspond. (Line 533-676)
Minor comments
Introduction
Point 7: Why did the authors decide to study this transcription factor and why they did it in Malus baccata?
Response 7: Thank you for your kind suggestion. In our previous study, the expression of MbNAC25 was significantly increased by transcriptome analysis under different stress treatments. This provides basis for further studying the function of this gene. We have improved the introduction in the manuscript to describe why we did the gene MbNAC25 and why we did it in Malus baccata. (Line 137-139)
However, the roles of NAC genes in Malus plant stress responses are less well known. Malus baccata is widely used as an apple rootstock or namental tree all over the world. It is highly resistant to low temperatures and drought [25]. In order to understand the role of NAC gene under low temperature and salt stress more comprehensively, a NAC transcription factor was isolated from the Malus baccata (L.) Borkh and named MbNAC25. The function of the transcription factor was analyzed and identified to provide new candidate genes for cold and salt tolerance breeding.
Point 8: Line 37. Number of NACs in tobacco is missing here.
Response 8: Thank you for your kind suggestion. We have read the reference "Tobacco Transcription Factors: Novel Insights into Transcriptional Regulation in the Solanaceae". In the study the author said "We found 203 complete or partial NAC domains in tobacco and a minimum number of 152 NAC genes.". So we change the sentence to "152 in soybean and at least 152 in tobacco " in line 40. (Line 40)
Point 9: Line 38. “NAC family genes form homologous or heterodimers at the N…” Please correct to homodimers if corresponds.
Response 9: Thank you for your kind suggestion. We have corrected the homologous to homodimers in line 41. (Line 41)
Point 10: Line 62. Authors mentioned TaNAC2 and TaNAC9, but don’t explain which species these transcription factors are from. Please mention.
Response 10: Thank you for your kind suggestion. We have added that TaNAC2 is from wheat in line 124. (Line 124)
Point 11: Line 63. What do the authors mean with “lower reaches of Arabidopsis”?
Response 11: Thank you for your kind suggestion. It is our mistake. We have moved “lower reaches of Arabidopsis” in line 125. (Line 125)
Results
Point 12: Line 73. How did the authors come up with this gene?
Response 12: Thank you for your kind suggestion. In our previous study, the expression of MbNAC25 was significantly increased by transcriptome analysis under different stress treatments. This provides basis for further studying the function of this gene. So we come up with this gene and find that over-expression of MbNAC25 in transgenic Arabidopsis plants enhances tolerance to cold and salinity stress.
Point 13: Line 82. What do the authors mean with “special sequences”?
Response 13: Thank you for your kind suggestion. It is our mistake. We have corrected the special sequences to conservative sequences in line 152. (Line 152)
Point 14: Line 86. Figure 1. The formatting of the letters A and B has to be corrected. Please, make them smaller and black.
Response 14: Thank you for your kind suggestion. We have corrected the letters A and B in line Figure 1. (Figure 1)
Point 15: Line 89. I suggest the authors write the names of all the species that they used for the analysis. They could write it here or in the materials and methods (M&M) section as they prefer.
Response 15: Thank you for your kind suggestion. We have written the names of all the species in the figure legend of Figure 1. (Figure 1)
Point 16: Line 102. Figure 2. The images lack scale bars, please add the bars and describe it also in the figure legend, such as “Scale bar corresponds to X um”.
Response 16: Thank you for your kind suggestion. We have added the bars in the images and describe them in the figure legend. (Figure 2 and Line 213)
Point 17: Line 116. Figure 3. I would suggest the authors use a greyscale for the bar graphs B and C since the pattern scale is more difficult to read.
Response 17: Thank you for your kind suggestion. We have changed the filled shape in bar graphs B and C in Figure 3. Tried our best to improve the figure and make it easy to read. (Figure 3)
Point 18: Figure 3. What do the authors mean with “Normal conditions”? Are they referring to “Control conditions”? I will recommend the authors use the term control conditions across the manuscript and that they describe those conditions in M&M. Regarding the figures 3B and 3C, what does CKstands for? The acronym is not explained.
Response 18: Thank you for your kind suggestion. We have changed the “Normal conditions” to “Control conditions” across the manuscript. CK in figures 3B and 3C stands for “Control conditions”. We have explained it in the figure legend of Figure 3 (Line 231)
Point 19: Line 119. What did the authors compare to do the statistical analysis? Could the authors mention it in the legend? For example: “Asterisks above columns indicate significant difference compared to what, in what condition? (*P≤0.05).”. Same should be addressed in the rest of the legends.
Response 19: Thank you for your kind suggestion. This is our negligence. We have addressed the statistical analysis in detail in the legends of Figure 3, 4, 5, 6, 7. (Line 230, 233, 266, 293, 307)
Point 20: Line 123-127. I suggest the authors move this paragraph to M&M. How they prepare the plants is not needed here.
Response 20: Thank you for your kind suggestion. We have removed this paragraph. We have described how to prepare the plants in the M&M. (Line 480-513)
Point 21: Line 124. Authors mentioned WT as an Arabidopsis line, but they did not explain which accession they are using for their experiments. This information has to be recorded.
Response 21: This is our negligence. The ecotype of WT Arabidopsis line is Columbia-0. We have added the accession of the WT Arabidopsis line in M&M. (Line 478)
Point 22: Line 135. The asterisks are not explained in the legend. There is no mention to the age of the plants in the experiment.
Response 22: Thank you for your kind suggestion. We have explained the asterisks in the legend. (Line 263-268) We have described the age of the plants in the M&M. (Line 479 and 503)
Point 23: Figure 4B. How do the authors measure the survival rate? It should be described in M&M
Response 23: This is our negligence. We have described how to measure the survival rate in the legend of Figure 4. (Line 263-266)
Point 24: What does CK in the axis mean? There is no explanation in the legend nor the M&M.
Response 24: Thank you for your kind suggestion. CK stands for “Control conditions”. We have explained it in the figure legend of Figure 4 (Line 263)
Point 25: Line 139. Figures 4B and 4C do not correspond with the text described in the legend, please modify accordingly.
Response 25: This is our negligence. We have modified the legend of Figure 4. (Line 260-268)
Point 26: Line 147. I would recommend starting the sentence with for example “Hence, our results suggest that overexpression of…”. This connects results and hypothesis.
Response 26: Thank you for your kind suggestion. We have accepted your suggestion to starting the sentence with “Hence, our results suggest that overexpression of…” in Line 274. (Line 274)
Point 27: Line 150. “Physiological Indices” please correct to physiological indexes.
Response 27: This is our negligence. We have corrected “Physiological Indices” to "physiological indexes". (Line 290)
Point 28: Line 153. Authors need to mention what they are comparing in the statistical analysis. Please modify across the text accordingly.
Response 28: Thank you for your kind suggestion. This is our negligence. We have addressed the statistical analysis in detail in the legends across the text. (Line 230, 233, 266, 293, 307)
Point 29: Line 156. “…were exposed to salt stress…” How do the authors perform NaCl treatment? I would suggest they use were watered if that is how they applied the stress.
Response 29: Thank you for your kind suggestion. We have changed the sentence to "the transgenic Arabidopsislines (S3, S6, S10) and WT line were watered by 200 mM NaCl for 7 days" and described the detals in the M&M. (Line 296, 480-513)
Discussion
Point 30: Line 192. I suggest the authors clarify this sentence, for example “… the highest homology with NAC25 in Malus domestica…”.
Response 30: Thank you for your kind suggestion. We have removed the sentence. We think this sentence is meaningless.
Point 31: Line 195. Do the authors mean Ohnishi et al.? Please modify the reference accordingly.
Response 31: Yes! It means Ohnishi et al. We have corrected the references across the manuscript to make them correspond. (Line 535-678)
Point 32: Line 196. “normal circumstances”, please explain.
Response 32: Thank you for your kind suggestion. This is our negligence. We have corrected it to "control condition". (Line 335)
Point 33: Line 204. I would suggest softening the statement such as “which could be due to the fact…”
Response 33: Thank you for your kind suggestion. We have accepted your suggestion to correcte it to “which could be due to the fact…”. (Line 355)
Point 34: Line 218. “This study” I suggest the authors modify this to “Our study” since helps the reader focus.
Response 34: Thank you for your kind suggestion. We have accepted your suggestion to correcte it to “Our study…”. (Line 372)
Point 35: Line 237. The last paragraph should be a summary with a conclusion of the work.
Response 35: Thank you for your kind suggestion. We have added a paragraph to summary the conclusion of the work in line 391-393. (Line 391-393)
Materials and methods
Point 36: Line 239. In the plant materials sections, there is no description of the plant materials used, for example, a description of Malus baccata variety, the Arabidopsis ecotype used, or the conditions Arabidopsis plants were grown.
Response 36: Thank you for your kind suggestion. The northeast dwarf apple (Malus baccata) is widely used as an apple rootstock in northern China. The Arabidopsis ecotype is Columbia-0. We have described the detals in the M&M. (Line 478)
Point 37: This paragraph could also include the treatments applied to transgenic Arabidopsis plants on soil, cold and salt stresses.
Response 37: Thank you for your kind suggestion. We have described the detals in the M&M.
Line 480-513:
Seeds of T2 generation of transgenic plants (S3, S6, S10, randomly selected) were sown and germinated on nutrient soil to vermiculite ratio is 4:1 in flowerpots (diameter 10 cm) with normal management in a growth chamber at 25 °C ± 1 °C under a 16 h light (50 μmol m-2 s-1)—8 h dark regime in parallel with wild-type (WT) seeds. 25 Seedlings were grown for 3 week with regular irrigation prior to salt stress. Salt stress experiments were conducted by watering 200 mM NaCl solution for 7 d. Then the 25 plants of each line (S3, S6, S10, and WT) were rewatered with water for 6 d to calculate the survival rate. The experiments were performed three times for each treatment at each time point. During the whole growth process, all Arabidopsis seedlings were observed and recorded by photographing on salt stress for 0 d, 7 d, and rewatered for 6 d (Recover). For low temperature stress, 3 week seedlings were grown in the 4℃ growth incubator parallel with wild-type (WT). The experiments were also performed three times for each treatment at each time point. During the whole growth process, all Arabidopsis seedlings were observed and recorded by photographing on cold stress for 0 h, 12 h, and recovered at room temperature for 24 h (Recover).
Point 38: Line 252. It will help clarify if authors mention for example “hydroponic seedlings of Malus baccata”.
Response 38: The hydroponic seedlings were cultured in Hoagland nutrient solution. When the leaves of seedlings were fully unfolded, the seedlings were for salt stress (watered with 200 mM NaCl solution), low temperature stress (placed in 4℃ growth incubator), drought stress (watered with 15% PEG solution) and high temperature stress (placed in 38℃growth incubator) treatment. (Line 448-450)
Reviewer 2 Report
The manuscript- ''Over-expression of a Malus baccata NAC Transcription Factor gene MbNAC25 Increases Cold and Salinity Tolerance in Arabidopsis'' by Han et al., presents the characterization of NAC transcription factor isolated from Malus baccata. The manuscript is well presented; however, I have a few the below-listed comments that the authors may find useful to improve the impact of their research work. 1-The abstract is fragmented. Each sentence sounds stand-alone and not connected with the rest of the text. For example, the first sentence-Abiotic stress plays an important role in the growth of most plants, is not followed by an appropriate background. I suggest the authors must provide the abstract in the following three sections- Introduction/Background to the research question, Methodology/Approach, and Results and Conclusion. The authors have tried this format; however, overall, the abstract reads fragmented. My suggestion would be- Abiotic stress plays a vital role in the growth of most plants. Plants respond to the stress regime via several different molecular mechanisms, including transcriptional regulation. Transcription factors are the key molecular switches orchestrating the growth–defense tradeoffs in plants.
2-Line 18- Mb should be MbNAC25
3-Line 21- The authors mentioned- Transgenic Arabidopsis was infected by Agrobacterium-mediated method. Did they use transgenic Arabidopsis? Or they used the Agrobacterium-mediated method to generate transgenic plants overexpressing the MBNAC25? 4-I will encourage the authors to re-write the introduction section as it reads very subjective. 5- The authors present the account of the function of NAC TF gene, however, throughout the manuscript they did not mention the NAC acronym which is derived from three proteins having a specific NAC domain i.e., NAM (no apical meristem), ATAF1/2, and CUC2 (cup-shaped cotyledon). 6-Line 69- Malus baccata should be italic 7-In Fig. 2, the authors mentioned MbWRKY1-GFP, I wondered whether it is a mistake or the localization for the WRKY1 instead of the NAC25. 8-Line 110-111, the authors described the expression pattern of the WRKY in Fig. 3B. 9- Figure 3. does not mention about the Fig 3 C. 10-The authors have analyzed several stress markers to understand MbNAC25 induced stress tolerance. However, they did not discuss how the NAC25 regulates these markers. And also, the proper referencing is required. For example, recently, the proline is shown to regulate the redox balance in Arabidopsis (please see- DOI: https://doi.org/10.1104/pp.16.01097). The authors have mentioned (line 214) that the SOD, CAT, POD, and MDA reduce the damage to plants by ROS scavenging. The authors should provide references for such statements, e.g., DOI:10.1002/9781119312994.apr0353). I would encourage the authors to discuss their findings comprehensively. 11- The authors should check the figure legends and numbers.
12-The manuscript lacks the overall summary or the conclusion. The authors must include the conclusion section.
Author Response
The manuscript- ''Over-expression of a Malus baccata NAC Transcription Factor gene MbNAC25 Increases Cold and Salinity Tolerance in Arabidopsis'' by Han et al., presents the characterization of NAC transcription factor isolated from Malus baccata. The manuscript is well presented; however, I have a few the below-listed comments that the authors may find useful to improve the impact of their research work.
Point 1: The abstract is fragmented. Each sentence sounds stand-alone and not connected with the rest of the text. For example, the first sentence-Abiotic stress plays an important role in the growth of most plants, is not followed by an appropriate background. I suggest the authors must provide the abstract in the following three sections- Introduction/Background to the research question, Methodology/Approach, and Results and Conclusion. The authors have tried this format; however, overall, the abstract reads fragmented. My suggestion would be- Abiotic stress plays a vital role in the growth of most plants. Plants respond to the stress regime via several different molecular mechanisms, including transcriptional regulation. Transcription factors are the key molecular switches orchestrating the growth–defense tradeoffs in plants.
Response 1: Thank you for your kind suggestion. We have re-write the abstract according to your opinion.
Point 2: Line 18- Mb should be MbNAC25
Response 2: This is our negligence. We have corrected it. (Line 260-268)
Point 3: Line 21- The authors mentioned- Transgenic Arabidopsis was infected by Agrobacterium-mediated method. Did they use transgenic Arabidopsis? Or they used the Agrobacterium-mediated method to generate transgenic plants overexpressing the MBNAC25?
Response 3: Thank you for your kind suggestion. We used the Agrobacterium - mediated method to generate transgenic plants overexpressing the MbNAC25. We have re-write the abstract and removed the sentence. (Line 13-28)
Point 4: I will encourage the authors to re-write the introduction section as it reads very subjective.
Response 4: Thank you for your kind suggestion. We have re-write the introduction, tried our best to improve the writing. If there are still errors, please give us another chance to correct them. (Line 31-142)
Point 5: The authors present the account of the function of NAC TF gene, however, throughout the manuscript they did not mention the NAC acronym which is derived from three proteins having a specific NAC domain i.e., NAM (no apical meristem), ATAF1/2, and CUC2 (cup-shaped cotyledon).
Response 5: Thank you for your kind suggestion. We have added explains of NAC in the abstract and introduction. (Line 13-14, 38).
Point 6: Line 69- Malus baccata should be italic
Response 6: This is our negligence. We have corrected it.
Point 7: In Fig. 2, the authors mentioned MbWRKY1-GFP, I wondered whether it is a mistake or the localization for the WRKY1 instead of the NAC25.
Response 7: This is our negligence. It is a mistake. We have corrected “MbWRKY1-GFP” to "MbNAC25-GFP ". (Line 211)
Point 8: Line 110-111, the authors described the expression pattern of the WRKY in Fig. 3B.
Response 8: Sorry! It is a mistake. We have corrected “MbWRKY1” to "MbNAC25 ". (Line 230)
Point 9: Figure 3. does not mention about the Fig 3 C.
Response 9: Sorry! This is our negligence. We have corrected it. (Line 228-233)
Point 10: The authors have analyzed several stress markers to understand MbNAC25 induced stress tolerance. However, they did not discuss how the NAC25 regulates these markers. And also, the proper referencing is required. For example, recently, the proline is shown to regulate the redox balance in Arabidopsis (please see- DOI: https://doi.org/10.1104/pp.16.01097). The authors have mentioned (line 214) that the SOD, CAT, POD, and MDA reduce the damage to plants by ROS scavenging. The authors should provide references for such statements, e.g., DOI:10.1002/9781119312994.apr0353). I would encourage the authors to discuss their findings comprehensively.
Response 10: Thank you for your kind suggestion. We have tried our best to improve the writing of the discussion, but the discusstion maybe not comprehensive due to the tight schedule. If the discussion still does not meet your requirements, please give us another chance to correct it. (Line 321-393)
Point 11: The authors should check the figure legends and numbers.
Response 11: Thank you for your kind suggestion. We have chected the figure legends and numbers across the manuscript and corrected the erros one by one.
Point 12: The manuscript lacks the overall summary or the conclusion. The authors must include the conclusion section.
Response 12: Thank you for your kind suggestion. We have added a paragraph to summary the conclusion of the work in line 391-393. (Line 391-393)